# OpenReview forum: "HybridSB-MoE: Dual-Domain Schrödinger Bridges with Scene-Adaptive Expert Routing for Speech Enhancement"
_ICLR.cc/2026/Conference — Submitted to ICLR 2026_

### Official Review · Reviewer_yA65 · 2025-10-27

**Soundness:** 2
**Presentation:** 1
**Contribution:** 2
**Rating:** 2
**Confidence:** 3

**Summary:**

This work proposes HybridSB-MoE, a dual-domain speech enhancement framework that integrates Schrödinger Bridge modeling with a heterogeneous Mixture-of-Experts for scene-adaptive processing. It reduces sampling steps from 40–50 to 8 while improving performance on VoiceBank+DEMAND.

**Strengths:**

- **Efficiency.** Compared with existing diffusion-based or bridge-based SE methods, the proposed HYBRIDSB-MOE framework is relatively efficient, requiring only 8 NFEs.
- **Performance.** The method achieves notable improvements on the VBD dataset, showing clear gains across multiple evaluation metrics compared to previous methods.

**Weaknesses:**

- **Writing.** The overall writing quality could be substantially improved. The paper lacks logical flow in the main text — frequent references to the appendix severely interrupt the reading experience. Moreover, the main body contains insufficient information: even after consulting the appendix, many crucial implementation details, such as model configuration and training parameters, remain unclear. In addition, several symbols and notations appear abruptly without proper definition or context, making it difficult to follow the technical formulation. Based on writing alone, I do not consider the paper ready for publication at ICLR; it would require a major revision to meet the conference’s readability and presentation standards.
- **Motivation.** Due to the weak writing and limited clarity (or perhaps my own reading limitations), the paper gives the impression of being a combination of multiple existing techniques—such as hybrid representation learning, scene-adaptive modeling, and uncertainty estimation—without clearly justifying their necessity or interdependence. The motivation for each design choice is not well explained. For example, it is unclear why uncertainty estimation is needed for a speech enhancement task, or whether the introduction of Mixture-of-Experts (MoE) is truly necessary in this context. Moreover, due to the lack of detail regarding the MoE implementation, I am unable to assess its actual role and contribution within the proposed framework.
- **Theory.** Due to the lack of sufficient contextual explanation and intermediate reasoning, verifying the correctness of the presented proofs was extremely difficult. Despite my best efforts, I was unable to reproduce or follow some of the key derivations, and several theorems—particularly those related to uncertainty estimation—appear more like definitions rather than rigorously proven results. Perhaps this is partly due to my limited mathematical background, but I would strongly encourage the authors to provide more detailed derivations and explanations to make the theoretical parts accessible to readers.
- **Efficiency.** Not all recent methods rely on 40–50 denoising steps. Some recent works, such as CDiffuSE and ROSE-CD, can achieve comparable or even better results in 1-6 steps. In addition, the paper lacks a fair comparison of model complexity (e.g., parameter count, FLOPs) and runtime efficiency (e.g., inference time per utterance) against baseline methods. Without these analyses, it is difficult to evaluate whether the reported performance gains stem from better modeling design or simply from increased model capacity and computational cost.

**Questions:**

- How do you compute ECE?

---

### Official Review · Reviewer_omDZ · 2025-10-29

**Soundness:** 3
**Presentation:** 3
**Contribution:** 3
**Rating:** 6
**Confidence:** 4

**Summary:**

Based on the field of generative speech enhancement, this paper aims to address the following issues: 1) current generative speech enhancement models are limited to single-domain processing, thus overlooking the complementary benefits of multi-domain representations demonstrated in separation tasks; 2) these models lack calibrated uncertainty quantification, which restricts their applicability across diverse scenarios; and 3) they suffer from high sampling costs.
The study designs a dual-branch network operating in the time-frequency domain. It integrates Schrödinger Bridges (SB) formulations with a heterogeneous mixture-of-experts for adaptive spectral processing under different acoustic conditions. By implementing trajectory regularization that combines optimal transport and path consistency, the model significantly reduces the number of sampling iterations required. Furthermore, comprehensive experiments are conducted on public datasets to validate the proposed approach.

**Strengths:**

（1）The model integrates both Schrödinger Bridges (SB) formulations and a heterogeneous mixture-of-experts approach to process speech information in parallel. This design not only leverages multi-domain information but also adapts dynamically to diverse acoustic scenarios.
（2）By incorporating the uncertainty from the learnable variance in the enhanced SB into calibrated fusion weights, it establishes a quantitative standard. Furthermore, a trajectory regularization method based on optimal transport is proposed. These innovations enable the model to accomplish the task with fewer sampling iterations, significantly reducing computational cost.

**Weaknesses:**

(1) Further elaboration on theoretical innovation: While this work primarily introduces innovations at the algorithmic level, it should be noted that Schrödinger Bridges (SB) formulations have already been applied in speech enhancement, and the heterogeneous mixture-of-experts is also an existing methodology. Therefore, it is essential to more explicitly clarify the specific contributions of this work to the SB formulations. Otherwise, the approach may be perceived merely as an "A+B" combination of two existing algorithms rather than a substantive theoretical advancement.
(2) Limited diversity in datasets: The VoiceBank+DEMAND dataset is characterized by high speech quality and well-maintained test set conditions, which may not comprehensively evaluate model robustness. It is worth considering the inclusion of additional public benchmarks, such as DNSChallenge, to assess model performance from multiple perspectives and under more varied acoustic scenarios.

**Questions:**

(1) We expect a further clarification on the algorithmic innovations, particularly in how this work advances beyond prior applications of Schrödinger Bridges in speech enhancement, and what novel modifications have been made to the heterogeneous mixture-of-experts framework. It would be valuable to see how the authors have specifically designed or adapted these algorithms to better align with speech-specific characteristics or to address the stated research problems.
(2) It is recommended to include experiments on additional datasets for comprehensive evaluation. Furthermore, experimental results on computational complexity should be supplemented, given the claim of significant energy efficiency improvement in the paper, which currently lacks supporting evidence.

---

### Official Review · Reviewer_r3FW · 2025-10-31

**Soundness:** 1
**Presentation:** 3
**Contribution:** 2
**Rating:** 2
**Confidence:** 4

**Summary:**

The paper proposes an hybrid architecture for speech enhancement employing Schroedinger bridge and mixture of experts.
The model has two branches, a schroedinger bridge branch that processes directly the waveform and another branch based on heterogeneous MoEs that instead is fed log-magnitude STFT representation.
The two predictions from the two branches are then fused using a uncertainty aware fusion step which fuses the waveforms by using the variance of the predictions between the MoEs and the variance of the SB generative process.

**Strengths:**

strong results on VoiceBank-DEMAND dataset.
The application of MoE in the speech enhancement field I think is new.

**Weaknesses:**

The main weakness of this paper is the rather limited empirical validation setup.
VoiceBank-DEMAND is a very small outdated dataset that has been around I think for a decade.
It is not very significant anymore when used alone since it is too small and many techniques that work well for this dataset then do not scale well or cannot tackle more complex acoustic conditions like joint noise and derverberation and more challenging and diverse noise conditions.
The noise conditions in VoiceBank DEMAND are rather limited in the diveristy and moreover the train, validation and test are reasonably matched domain.

The authors should consider adding experiments on more up to date datasets considering for example WHAMR!, CHiME 4, DNS challenge, URGENT challenge or CHiME 7 UDASE  where the systems need to generalize from simulated to real world scenarios.

The use of MoEs also is called into question in such a very small limited data setup and may not bring many benefits.
In fact In Figure 2 sometimes the performance has very little variation  (e.g. PESQ 3.84 to 3.92 which may be very well be between the confidence interval looking at Table 2).
I think the paper would benefit by some additional ablation studies on if we actually need the MoE architecture at all, or the improvement is simply because it is just a parallel supervised predictive branch.
I think these baseline systems should be added:
1.  manual and random expert routing
2. a single network with no MoE but similar number of parameters is used.
3. homogeneous MoE to justify the heterogeneous choice.

Some of the presented theory (Theorem 1 ) is already well established or not needed or trivial (e.g. the heterogeneous universal approximation). Theorem 10 it appears to me just standard variance decomposition.
I think that the paper is unnecessary heavy on math.
It would be beneficial to move these in the Appendix and add instead more empirical validation and ablations which are more critical.

**Questions:**

I do not have particular questions.

---

### Official Review · Reviewer_S2p3 · 2025-11-01

**Soundness:** 2
**Presentation:** 3
**Contribution:** 3
**Rating:** 4
**Confidence:** 2

**Summary:**

This paper proposes HybridSB-MoE, a dual-domain speech enhancement framework that runs in parallel a waveform-domain enhanced Schrödinger Bridge (SB) denoiser and a heterogeneous mixture-of-experts (MoE) spectrogram pathway, and then fuses the two outputs using uncertainty-aware weights. The SB side introduces trajectory regularization (optimal transport + path consistency) to reduce sampling steps (claim: 8 steps vs 40–50) while the MoE side routes to scene-specialized experts with sparse Top-k routing. On VoiceBank+DEMAND, the paper reports PESQ 3.88±0.25 and STOI 0.96, with ablations attributing notable drops when removing either SB or MoE.

**Strengths:**

1. Clear, well-motivated architecture synthesis. The dual-domain design is a novel design which is addresses complementary strengths: spectral branch for harmonic structure and waveform SB for phase/temporal coherence, combined via uncertainty-aware fusion.

2. Efficiency focus with few-step SB. The trajectory regularization and adaptive timesteps are argued to enable K=8 sampling with maintained quality, a practical step toward real-time use where prior diffusion/SB works often need 40–50 steps.

3. Competitive results on a standard benchmark. The reported PESQ 3.88±0.25 / STOI 0.96 are competitive; the table compares against relevant recent baselines (e.g., SGMSE-style, SBCTM, ROSE-CD, Mamba-based).

**Weaknesses:**

1. Internal inconsistency on the number of experts. The abstract and text repeatedly state five scene categories (Home/Nature/Office/Transport/Public), yet Figure 1 caption says “14 scene-specific experts with Top-k=2.”

2. Limited evaluation breadth. Results are only on VoiceBank+DEMAND. Given claims of scene adaptivity and robustness, additional corpora or noisy/reverb sets (DNS Challenge, WHAM!/WHAMR!, CHiME) would strengthen generalization claims; otherwise, the MoE’s scene specialization risks overfitting to the specific scene taxonomy of VoiceBank+DEMAND. (The paper does show a scene-stratified analysis figure, but all within the same dataset.)

3. Comparative fairness on step counts. The paper positions 8-step inference against prior works typically using 40–50 steps, but use diffusion models for comparison. Recent SB works (e.g. https://arxiv.org/pdf/2407.16074) states that about 10 sampling steps suffice for high-quality SE with SB.

4. Theoretical claims need sharpening. The paper made a wide range of theoretical statements (convergence, error bounds), but these are not fully integrated into the empirical narrative. Several Theorems need strong assumptions and omit proof in the main text. More experimental results validating the theoretical predictions would help close the theory-experiment loop.

5. Ablations could be more detailed. Beyond “w/o SB / w/o MoE / sequential,” it would help to ablate (i) OT vs path-consistency components separately, (ii) Top-k value and capacity factor in load balancing, and (iii) number/architectures of experts to support the heterogeneous-MoE design choice.

**Questions:**

See Weaknesses

---

### Meta-Review · Area_Chair_ExuP · 2026-01-06

**Summary:**

The paper has several issues, including limited evaluation on VoiceBank and DEMAND (raised by reviewer S2p3 and r3FW), limited novelty (raised by reviewer r3FW and omDZ), and unclear writing (raised by reviewer S2p3, yA65).

Fixing these issues would require a substantial revision to the submission.

**Reviewer Concerns:**

No rebuttal submitted.

**Reviewer Scores:**

No rebuttal submitted.

---

### Decision · Program_Chairs · 2026-01-26

Reject